# Experimental Study, Simulation and Analysis of the Fracture Failure of the Drum Shaft of a Casting Bridge Crane

**Dong Xiang [1],*, Yan Li [2], Yuqing Zhang [2] and Feng Xu [1]**

1    The School of Mechanical Engineering, University of Science and Technology Beijing, Beijing 100083, China
2    China United Engineering Corporation Limited, Hangzhou 310052, China
*    Correspondence: xdustb@163.com; Tel.: +86-010-62332538

**Abstract:** This study investigated the fatigue fracture of bilateral drive drum shafts in casting bridge cranes including its fracture morphology and factors, such as materials, manufacturing processes, and loads. Seven conditions were designed to test the effects of changes in the speed and torque of the drum shafts during startup, commissioning and braking under different loads. A dynamic model was developed for the structure and control system of the hoisting mechanism. Changes in the speed and torque of the motor and drum shafts were simulated under common operating conditions such as speed and load changes of the motor, control asynchrony and single-motor towing. The results showed that asynchronous motor starting and braking, motor dragging and other behaviors led the left and right drum shafts undergo oscillated torque with a value reached $2 \times 10^5$ N·m in a period of approximately 13 s, and a residual torque about $3 \times 10^4$ N·m was retained after braking. The torques on the drum shafts changed suddenly during the processes of starting, shifting and braking. Dynamic loading was the root cause of fatigue fracture of the drum shafts.

**Keywords:** crane; drum shaft; fatigue; fracture

## 1. Introduction

Fatigue fracture, which accounts for more than 50% of the fractures of mechanical parts [1], has long been the focus of fracture failure research. Recent studies have considered many factors that contribute to fatigue fracture [2–11], such as materials, structure, manufacturing processes, service conditions, and dynamic behavior.

Fatigue fracture of the core components used for the transmission of rotation and torque in the drive shaft is very dangerous to machines and humans. The motor shaft was the object of the world's first fatigue testing machine [12]. Currently, research on fatigue fracture of drive shafts has been conducted for various machinery and equipment, such as automobiles, construction machinery, rail vehicles, and air compressors. [13–18] However, the causes of fatigue have not been related to the differences in materials, structure, and manufacturing processes, leading to continuous research effort focusing on different service conditions and equipment.

Casting bridge cranes, which are mainly used to lift molten iron and molten steel, have the characteristics of large lifting capacity, high operation rate and harsh working conditions. Once fatigue fracture occurs hereby, it will result extremely dangerous accidents. After an accident, in which a ladle of molten steel overturned at Qinghe Special Steel Co., Ltd. in 2007, casting bridge cranes were required to use a bilateral main hoisting mechanism for greater safety [19].

The bilateral propelled hoisting mechanism contains two load drums with shafts at their ends. One load drum shaft is connected to the output shaft of the reducer through a coupling, and the other load drum shaft is connected to the other load drum through a coupling too. The load drum, as a component in the hoisting mechanism that experiences very complex forces, carries the dynamic load produced not only by the lifting and dropping

of the workpiece but also by the operation, control and behavior of the motor, transmission system, braking system and other components. Thus, failure of load drums is responsible for approximately 25% [20], the largest proportion, of crane accidents. The load drum shaft is the part of the load drum that is most likely to fail. More importantly, once the load drum shaft breaks, it causes downtime and even serious accidents due to falling objects. For example, at Hebei Tangshan Steel Co., Ltd., a serious accident occurred in which the load drum shaft broke when the casting crane was lifting a ladle of molten iron. The coupling that connected the load drum and the reducer tore, and the load drum fell off the frame; accordingly, the ladle was inclined and molten iron overflowed, and consequently, the production line burned [21]. Although the failure of load drum shafts has always been a concern of the crane industry, current research into load drum shaft failure is still mainly based on qualitative analysis [22–26].

Taking the failure of a 50/10 t casting bridge crane load drum shaft as an example, a testing and analysis of dynamic factors in the fatigue fracture of the load drum shaft in the hoisting mechanism are carried out after a failure factor investigation and qualitative analysis considering typical working conditions. The sources of the loads are clarified, and their dynamic responses are quantified. As a result, design accuracy and operational control must be improved.

## 2. Investigation of the Reasons for Fractures

Fracture failure usually involves a combination of multiple factors. To determine the causes of a fracture, the possible factors are investigated, such as fracture appearance, materials, manufacturing processes, loads, etc.

### 2.1. Fracture Appearance Analysis

A fracture of a load drum shaft is shown in Figure 1. Unfortunately, the fracture features were not completely preserved because of friction. However, the cracks nearer to the shaft surface are smoother, and the cross-section does not have obvious necking, and the cracks are inclined in one direction with a large depth inside. In addition, as shown in Figure 1b, the source of the crack in the drum shaft is near the surface of the shaft, the final crack area is located inside the shaft, and scallops are located between the two areas. Therefore, the fracture was caused by rotational bending fatigue; the load drum shaft experienced a large torsional force prior to fractured.

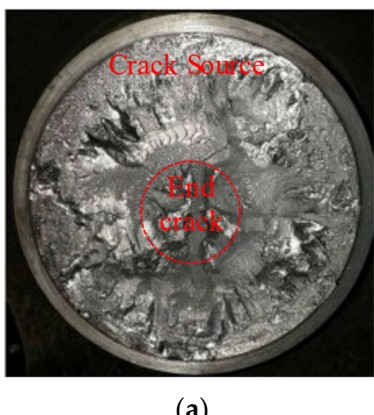
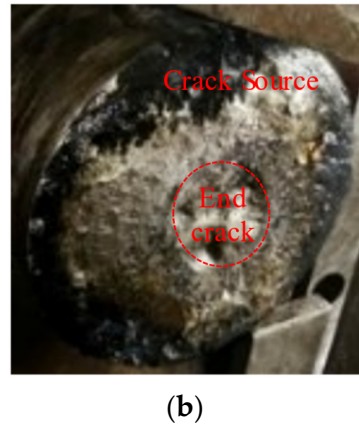

(**a**)             (**b**)

**Figure 1.** Appearance of the fracture of a load drum shaft: (**a**) cross-section of the left load drum shaft; (**b**) cross-section of the right load drum shaft.

The load drum shaft broke at the neck journal root with diameter Φ220 mm, and a magnified image of the transition fillet is shown in Figure 2. According to the standards in *Parts Rounding and Chamfering*, when the shaft shoulder diameter difference is more than 40 mm, the fillet radius should be 4 mm. However, the actual fillet radius of this shaft shoulder was approximately 1.5 mm, which may have led to a high stress concentration.

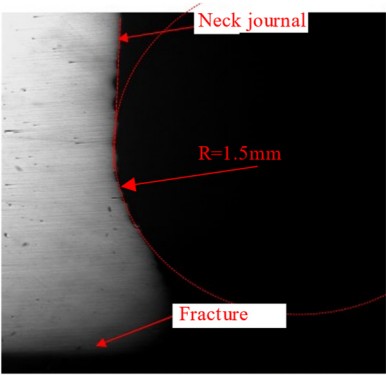

**Figure 2.** Transition fillets of the drum shaft, 20X.

*2.2. Material Analysis*

According to the design requirements, the material of the failed load drum shaft should be #35 steel, and it must be normalized and tempered, so that its hardness reaches 137–163 HBW. After testing, the hardness of this load drum shaft was distributed in the range of 153–161 HBW, and the core hardness was lower than the surface hardness, satisfying the technical requirements.

Samples were drilled respectively at the spaces of near the surface, 1/2 radius and core of the shaft for analysis of the chemical composition. As shown in Table 1, the material composition met the requirements for #35 steel in manufacturing standard *GB/T699*. From the metallographic structure shown in Figure 3, the structure of the shaft was normalized pearlite and ferrite with few inclusions, which also meeting the performance requirements of the spool shaft material.

**Table 1.** Chemical Composition of the Drum Shaft, w%.

| Elements \ Location | Surface | 1/2R | Core | Required |
|---|---|---|---|---|
| C | 0.37 | 0.34 | 0.35 | 0.32–0.39 |
| Mn | 0.59 | 0.58 | 0.59 | 0.50–0.80 |
| Si | 0.22 | 0.22 | 0.22 | 0.17–0.37 |
| P | 0.018 | 0.017 | 0.017 | ≤0.035 |
| S | 0.002 | 0.002 | 0.002 | ≤0.035 |
| Cr | 0.01 | 0.01 | 0.01 | ≤0.10 |
| Ni | 0.01 | 0.01 | 0.01 | ≤0.30 |

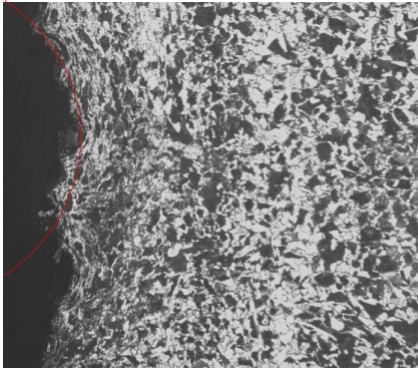

**Figure 3.** Microstructure of a cross-section, 100X.

*2.3. Manufacturing Process Analysis*

In the processing of the load drum and the load drum shaft, manufacturing and acceptance were carried out according to the manufacturing standard *DHQ. JS001*, cir-

cumferential and longitudinal welds were 100% UT inspected, and the quality reached BI grade and BII grade in manufacturing standard *GB/T11345*. Short shaft and plate welds were 100% MT inspected 48 h after welding, and the quality reached level 2 according to manufacturing standard *JB/T6061–2007*. In addition, a total annealing treatment after welding was carried out to improve the tissue defects and residual stress of the load drum. Therefore, the influence of the manufacturing process on the fracture failure can be ignored.

### 2.4. Service Condition Analysis

Table 2 shows the working level of the crane and its hoisting mechanism with a broken load drum shaft. The working level means that the crane frequently withstands cyclic loads when it is operated, which may lead to fatigue.

**Table 2.** Working rank of the crane and its lifting mechanism.

| Items | Work Level | Use Level | Loads Level |
|---|---|---|---|
| Crane | A7 | U5 | Q4 |
| Hoisting mechanism | M8 | T7 | L4 |

### 2.5. Static Analysis of the Load Drum Group

As shown in Figure 4, the load drum group of the bilateral propelled hoisting mechanism contained two load drums. Both ends of the two load drums were load drum shafts, one load drum shaft was connected to the output shaft of the reducer, and the other load drum shaft was connected to the other load drum through a GIICL type drum gear coupling. The load drum group had four supports. The left and right ends were supported by the reducer box, and the load drum shafts (in addition to the coupling) were supported by the bearing pedestal.

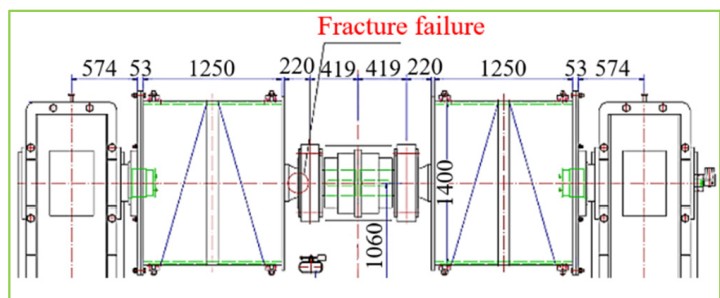

**Figure 4.** Bilateral drive drum.

The simplified static model of the bilateral drive load drum is shown in Figure 5. The values and meanings of the codes in Figure 5 are shown in Table 3. The following assumptions were made:

1. The coupling connecting the left and right load drum shafts can transmit torque without loss, and it was simplified to a hinged connection.
2. The axial movement of the load drum group was ignored, and the bearing pedestal of the two load drum shafts was converted into a hinge support.
3. The weight of the load drum and the coupling were uniform loads, the wire rope tension was the concentrated force acting on the center of the load drum, and the quality of the load drum shaft was ignored.

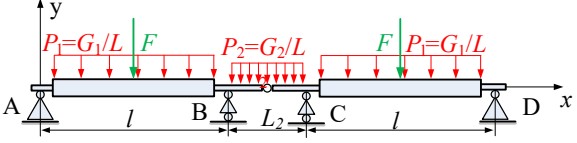

**Figure 5.** Simplified static model of the bilateral drive drum.

**Table 3.** Values and meanings of the codes in Figure 5.

| Codes | Values |
|---|---|
| Load drum length $L$/mm | 1331 |
| Load drum shaft length $L_1$/mm | 220 |
| Coupling length $L_2$/mm | 838 |
| Load drum group quality $G_1$/kN | 44 |
| Coupling quality $G_2$/kN | 8 |
| Unilateral lifting weight $G_3$/kN | 250 |
| Pulley set magnification $N$ | 2 |
| Single load drum load $F$/kN | 125 |

Considering the above assumptions and the symmetrical structure of the couplings, the force and flexural moment of the equivalent hinge of the coupling were both 0, and the static model in Figure 5 was simplified to the beam structure shown in Figure 6a. The gear engagement of coupling generated a bending moment Ma in the axial direction due to friction. According to *JB/T 8854.2-2001*, $M_a = 0.1T_{max}$, where $T_{max}$ is the maximum torque acting on the coupling. Obtained by calculations using the mechanics theory of materials science, the shearing force diagram and bending moment diagram are shown in Figure 6b,c, and the maximum absolute value of the shearing force and bending moment for the load drum and load drum shaft were calculated using Equations (1)–(3).

$$\begin{aligned}|F_q|_{max1} &= |F_q|_{max2} = \left|-F_{By} + \frac{G_2}{2}\right| \\ &= \left|-\frac{F+G_1}{2} - \frac{1}{l}\left(M_a + \frac{G_2L_2}{8}\right)\right|\end{aligned} \tag{1}$$

$$\begin{aligned}|M|_{max1} &= M^+_{max} = \frac{F_{Ay}l}{2} - \frac{G_1L_1}{8} \\ &= \frac{(F+G_1)l}{4} - \frac{1}{2}\left(M_a + \frac{G_1L_1}{4} + \frac{G_2L_2}{8}\right)\end{aligned} \tag{2}$$

$$|M|_{max2} = M^-_{max} = \left|-\left(M_a + \frac{G_2L_2}{8}\right)\right| \tag{3}$$

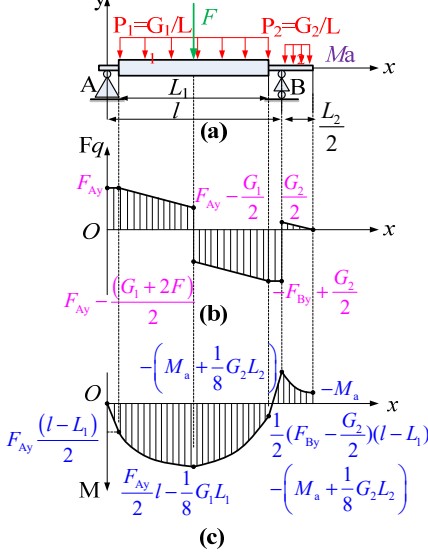

**Figure 6.** The force and flexural moment analysis of drum: (**a**) static model of the bilateral drive drum; (**b**) the shear force diagram; (**c**) bending moment diagram.

After calculations using Hooke's law and the moment deformation equation of the simply supported beams, the maximum normal bending stress of the load drum body section and load drum shaft section were written respectively as follows.

$$\sigma_{max1} = \frac{|M|_{max1}}{W_1} = \frac{M_{max1}D}{2I_1} \tag{4}$$

$$\sigma_{max2} = \frac{|M|_{max2}}{W_2} = \frac{M_{max2}d_2}{2I_2} \tag{5}$$

where $\sigma_{max1}$ is the maximum normal bending stress of the load drum body section, $\sigma_{max2}$ is the maximum normal bending stress of the load drum shaft section, $I_1$ is the cross-sectional moment of inertia of the load drum body, $D$ is the outside diameter of the load drum body, $d_1$ is the inside diameter of the load drum body, $d_2$ is the diameter of the load drum shaft, and $I_2$ is the cross-sectional moment of inertia of the load drum shaft.

If the cross-sectional maximum shearing stress of the load drum body and load drum shaft are both distributed on the neutral axis, then the maximum bending shear stress of the load drum body and load drum shaft can be written as:

$$\tau_{max1} = 16\frac{|Fq|_{max1}}{\pi(D^2 - d_2^2)} \tag{6}$$

$$\tau_{max2} = \frac{16}{3}\frac{|Fq|_{max2}}{\pi d_1^2} \tag{7}$$

Because $T_{max}$ is small in the load drum group of the bilateral propelled hoisting mechanism, $M_a$ can be ignored. After calculation, the values of $\sigma_{max1}$, $\tau_{max1}$, $\sigma_{max2}$, $\tau_{max2}$ are 1.7 MPa, 2.2 MPa, 5.2 MPa, 7.75 MPa, respectively. All of these values are far less than the ultimate strength of the load drum and load drum shaft, which are 235 MPa and 315 MPa, respectively.

### 2.6. Dynamic Load Analysis of the Hoisting Mechanism

As shown in Figure 7, when the hoisting mechanism operating, the motor drives the transmission components, such as the reducer, the load drum, and the wire rope, to lift heavy objects. The hoisting mechanism bears dynamic loads that include the dynamic electromagnetic torque generated by the motor under the speed control system, and the loads, which are the forces generated by the structure with friction, assembly tolerance, interstice and damping behavior, and the forces generated by weight while lifting, lowering and braking. Particularly for the bilateral propelled lifting mechanism, high dynamic loads are carried for the asymmetric structure, and the nonsynchronous control system changes the force distribution in its driving system.

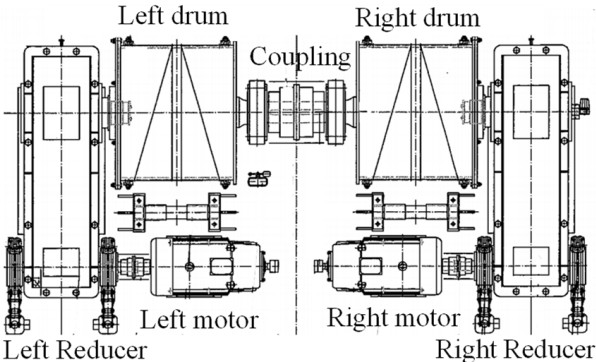

**Figure 7.** Structural layout diagram of the lifting mechanism.

## 3. Dynamic Test of the Lifting Mechanism

### 3.1. Test System Setup

To study the dynamic behavior of the hoisting mechanism, a test system was constructed on a cast bridge crane with a broken drum shaft, as shown in Figure 8. The test system included a strain-torque telemetry equipment, strain acquisition sensor, data acquisition equipment and other modules. Table 4 summarizes the performance parameters of the cast bridge crane, and Table 5 summarizes the types of main instruments for the test.

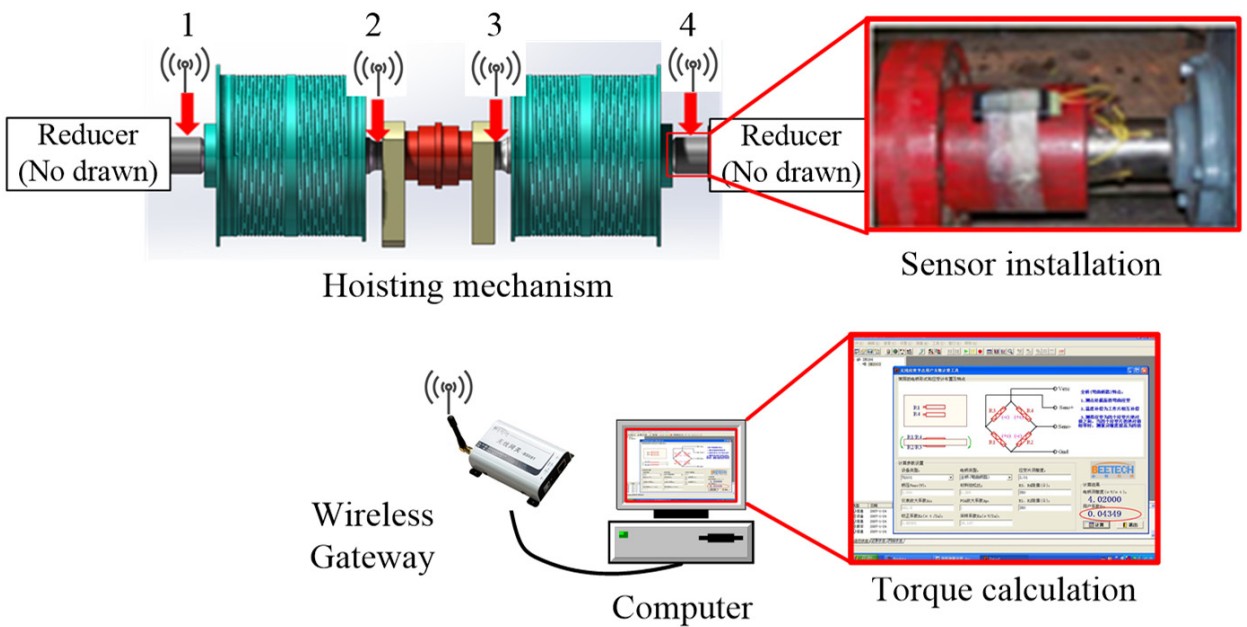

**Figure 8.** Composition of the dynamic test system for the lifting mechanism.

**Table 4.** Main parameters of a casting bridge crane.

| Properties | Lifting Capacity | Length | Beam Weight | Trolley Weight | Motor Power | Rated Speed |
| --- | --- | --- | --- | --- | --- | --- |
| Units | t | m | t | t | kW | rpm |
| Values | 50 | 22.5 | 183 | 29 | 75 | 750 |

**Table 5.** Main instruments for testing.

| No. | Instrument Model | Application |
| --- | --- | --- |
| 1 | KFW-2-120-D16-11 L1M2S | Strain test |
| 2 | TQ201 No. 2126 | Torque and speed test |
| 3 | BS903 | Wireless receiving gateway |
| 4 | BeeData | Software for signal acquisition and processing |

To compare the dynamic behavior under different working conditions, 7 kinds of working conditions, listed in Table 6, were tested. The speeds in four gears of the crane during test are listed in Table 7. In each working condition, the rotational speed and stator current of the motor and the torque of the load drum shaft were measured; the detection positions were named positions 1, 2, 3 and 4, as shown in Figure 8.

**Table 6.** Working conditions for the tests and simulations.

| Type | Object Weight | Speed and Direction | Operation Method |
|------|---------------|---------------------|------------------|
| Type 1 | None | Static | The hook is static without weights, and the sensor is set to zero |
| Type 2 | None | 4th gear and dropping | After the hook falls a certain vertical distance, quickly decelerate from the 4th gear to the 1st gear and finally stop in the air |
| Type 3 | | 4th gear and lifting | After the hook lifts a certain vertical distance, quickly decelerate from the 4th gear to the 1st gear and finally stop in the air |
| Type 4 | 40.44 t | 4th gear and lifting | After the object is lifted a certain vertical distance, quickly decelerate from the 4th gear to the 1st gear and finally stop in the air |
| Type 5 | | 4th gear and dropping | After the object falls a certain vertical distance, quickly decelerate from the 4th gear to the 1st gear and finally stop in the air |
| Type 6 | ≤40.44 t | 4th gear and dropping or lifting | The heavy object is always placed on the ground, and the hoisting mechanism repeatedly lifts and descends in 1 block |
| Type 7 | 40.44 t | 4th gear and dropping | After the object falls a certain vertical distance, quickly decelerate from the 4th gear to the 1st gear and finally stop on the ground |

**Table 7.** Gears and speeds.

| Gear | Percentage of Maximum Speed | Motor Speed | Load Drum Speed |
|------|------------------------------|-------------|-----------------|
| Units | % | r/min | r/min |
| 4 | 100 | 745 | 4.3 |
| 3 | 30 | 223.5 | 1.29 |
| 2 | 20 | 149 | 0.86 |
| 1 | 10 | 74.5 | 0.43 |

*3.2. Test Results Analysis*

Since working condition 6 is most frequently used, and the dynamic behavior of the crane in this condition is most representative for it involves operations such as lifting, dropping, braking and direction adjustment, working condition 6 was taken as an example for analysis. The motor speed, gear position and torque of the load drum shaft for 30 s are shown in Figure 9.

Figure 9a illustrates the motor speed and gear of the hoisting mechanism when it repeatedly lifts, drops and brakes with objects on the ground. Figure 9b illustrates the torques in positions 2 and 3. It can be concluded that the torque on the load drum shaft changed suddenly due to the change in the hoisting weight when the gear changed from the zero point and the motor accelerated the lifting object. Additionally, the torque on the load drum shaft changed suddenly during braking and then stabilized at a constant value, just like it was affected by the step signal. It is clear that frequent starting and braking brought periodic reciprocating vibrations and load changes for the transmission system. Otherwise, as shown in Figure 9a, the motor speed lagged behind the gear change, and the motor first reversed and then rotated forward due to insufficient lifting torque at low gear—that is, it was dragged by heavy objects and in a state of power generation.

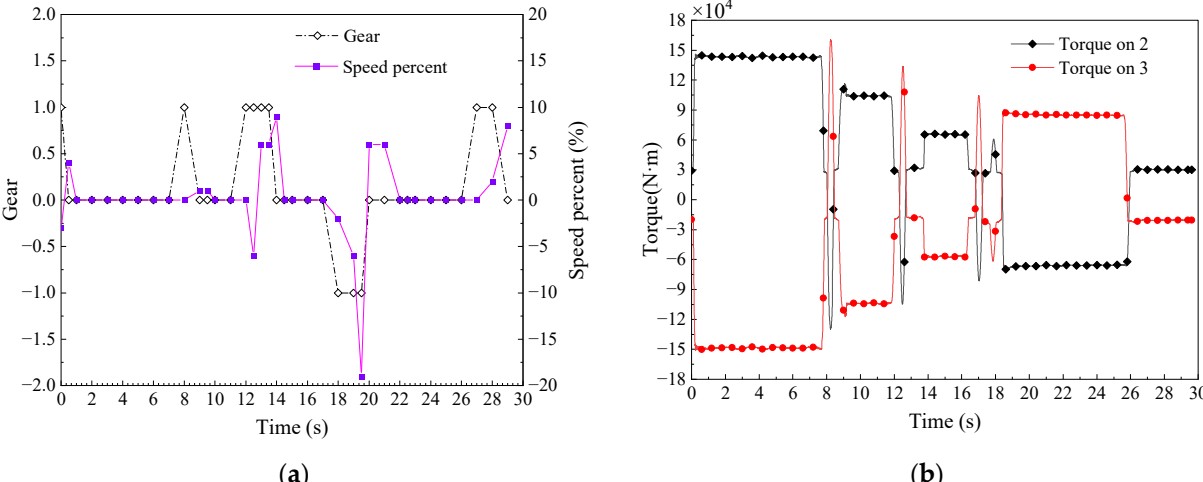

**Figure 9.** Crane lifting weight fast and frequent starting-braking test: (**a**) The motor speed and gear of the hoisting mechanism; (**b**) frequent starting-braking drum shaft torque change.

The dynamic behavior of the crane under other working conditions was similar to that under working condition 6. Once the gear position changed, lifting, falling or braking would consequently occur, and vibrations and impacts would occur in the hoisting mechanism. When objects rose or fell, there was a certain phase difference between the torques on the two load drum shafts, as shown in Figure 10.

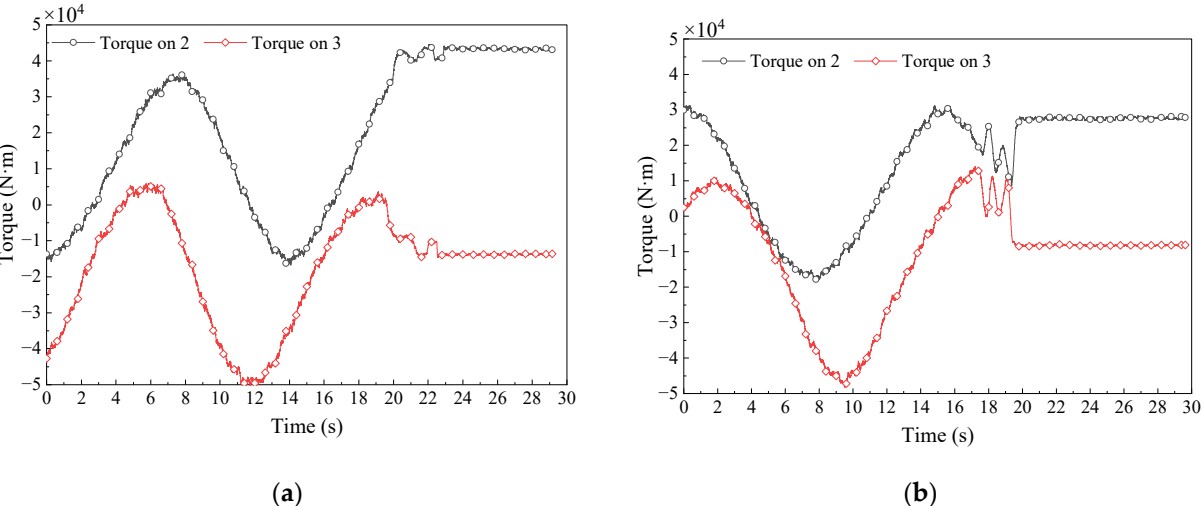

**Figure 10.** Torque change of the drum shaft before and after braking of the bilateral drive lifting mechanism: (**a**) lifting brake in 1st gear when lifting 40.44 t; (**b**) lowering brake in 1st gear when lifting 40.44 t.

Figure 10a,b show the torques measured at points 2 and 3 when the crane worked in the 1st gear, and the object weight was 40.44 t. All braking started at 20 s. Prior to braking, when the object was being lifted or dropped, the torques on the load drum shafts vibrated with a high amplitude and low frequency, and there was a phase difference between them. After braking, the torques on the load drum shafts remained stable after short-term vibrations at high frequency.

The test results showed that the service conditions strongly influenced the dynamic performance of the hoisting mechanism, and the torque on both load drum shafts of the hoisting mechanism changed periodically; the period was approximately 13 s. Otherwise, after braking, the torque values of the two load drum shafts changed abruptly, but the

changes were in opposite directions. The analysis showed that this may be related to motor drag and asynchronous brake holding in the bilateral drive hoisting mechanism.

For an in-depth explanation of the behavior observed in the above-described test and improve the design of the spool shaft, a dynamic model was constructed and the hoisting mechanism was simulated.

### 4. Electromechanical Coupling Dynamics Model of the Bilateral Propelled Hoisting Mechanism

#### 4.1. Dynamic Simulation Process

As shown in Figure 11, the hoisting mechanism is divided into three modules: the motor and its control, the mechanical transmission system and hoisting weight. Each modules contain its own structural parameters, control parameters, and working parameters, and there are dynamic loads, motion parameters, and control parameters to realize data transmission and feedback between them. Therefore, it is necessary to build an electromechanical coupling dynamics model of the motor, its control system and the mechanical transmission system, in order to analyze the dynamic response of the hoisting mechanism.

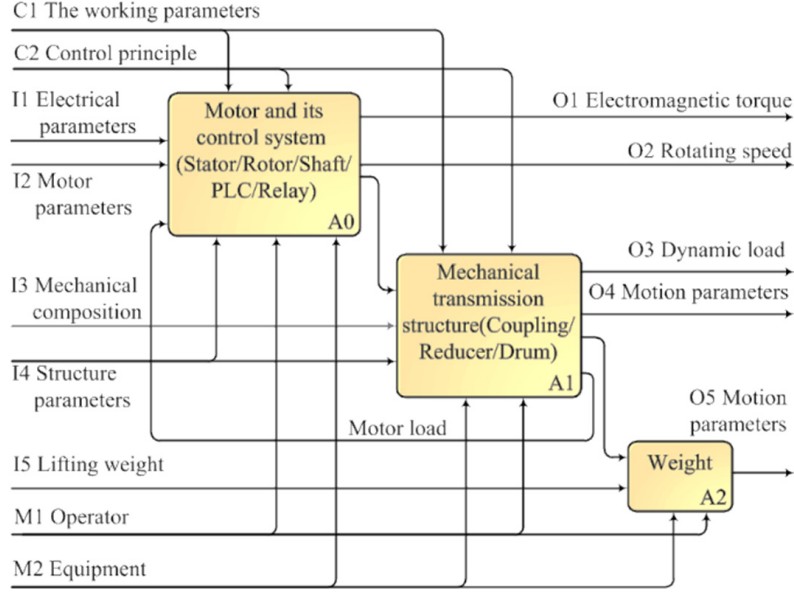

**Figure 11.** Structural layout diagram of the hoisting mechanism.

However, it is difficult to carry out a dynamic simulation if the electromechanical model is too complex. Therefore, the following assumptions are made to simplify the model.

1.  The vibration of the system in the horizontal plane and the swing of the weight are ignored, and only the vertical motion and the torsion of the structure are considered.
2.  The friction between the wire rope and the drum, and the rigid resistance of the wire rope as well are ignored.
3.  The contact stiffness at the connection for the reducer and the drum, which are rigidly connected, is ignored. That is, only the torque stiffness and torsional damping of the connecting shaft are considered.
4.  The torsional deformation of the reducer, drum and coupling are ignored.

Based on the above assumptions, considering the electromechanical coupling effect and simplifying the transmission components into a mechanical equivalent model composed of mass, stiffness, and damping, as shown in Figure 12, the motor, its control system and transmission mechanicals are also considered. The relevant parameters in the vertical translation-torsion dynamic equivalent model are shown in Table 8.

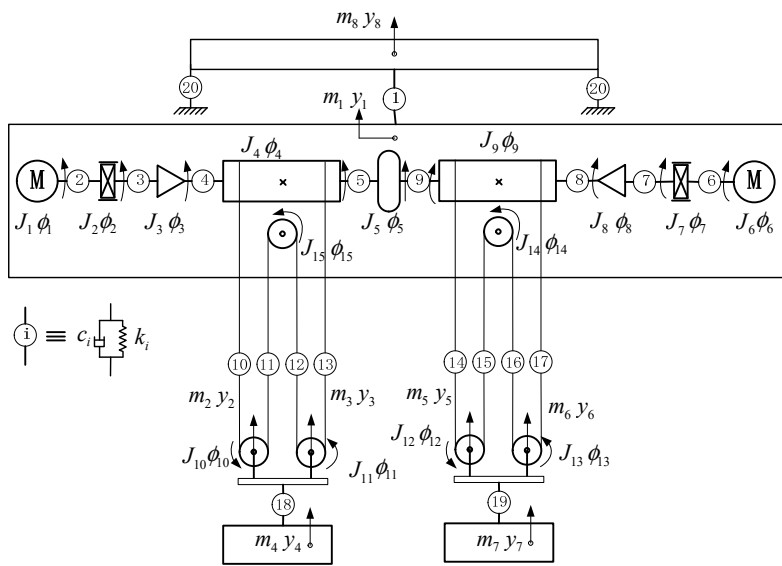

**Figure 12.** Vertical translational-torsional dynamic model of the bilaterally driven lifting mechanism.

**Table 8.** The parameters for the bilateral driven crane mechanisms and their meanings are shown in Figure 12.

| Parameters | Object Weight | Parameters | Operation Method |
|---|---|---|---|
| $m_1$ | Equivalent weight of the hoisting mechanism | $m_2 + m_3, m_5 + m_6$ | Weight of the hooks |
| $m_4, m_7$ | Weight of the objects | $m_8$ | Weight of the crane girder |
| $J_1, J_6$ | Moment of the motor | $J_2, J_7$ | Moment of the brake |
| $J_3, J_8$ | Moment of the reducer | $J_4, J_9$ | Moment of the load drum |
| $J_5$ | Moment of the coupling | $J_{10} \sim J_{15}$ | Moment of the pulley block |
| $k_i$ $i = 1,10,11,12\ldots 20$ | Translational stiffness coefficient | $k_i$ $i = 2,3,\ldots 9$ | Rotational stiffness coefficient |
| $c_i$ $i = 1,10,11,12\ldots 20$ | Translation damping coefficient | $c_i$ $i = 2,3,\ldots 9$ | Rotational damping coefficient |
| $r_4, r_9$ | Radius of the load drum | $r_i$ $i = 10,11,\ldots 15$ | Radius of the pulley |
| $y_i$ $i = 1,2\ldots 8$ | Displacement of the crane masses | $\theta_i$ $i = 1,2,3\ldots 15$ | Rotation angle of the crane parts |

Since this research mainly focuses on the influence of the bilateral driving arrangement on the dynamic load distribution in the hoisting mechanism, it is assumed that the structure is symmetrical. That is, the motors, brakes, reducers, drum sets, pulley sets and hook sets on both sides are considered to be identical, and the manufacturing errors and assembly errors of individual components are ignored. In addition, it is assumed that the stiffness and damping of each wire rope in the unilateral pulley block are the same. The relationship between the parameters in Figure 12 are given in Table 9.

**Table 9.** Relationship between the parameters in Table 7.

| Number | Name | Relationship |
|---|---|---|
| 1 | Weight | $m_2 = m_3 = m_5 = m_6 = m_h, m_2 + m_3 = m_5 + m_6 = m_j$ |
| 2 | Displacement | $y_2 = y_3 = y_1, y_5 = y_6 = y_r$ |
| 3 | Radius | $r_{10} = r_{11} = r_{12} = r_{13} = r_{14} = r_{15} = r_h$ |
| 4 | Moment | $J_1 = J_6, J_2 = J_7, J_3 = J_8, J_4 = J_9$ |
| 5 | Stiffness | $J_{10} = J_{11} = J_{12} = J_{13} = J_{14} = J_{15} = J_h$ |
| 6 | Damping | $k_2 = k_6, k_3 = k_7, k_4 = k_8, k_5 = k_9, k_{18} = k_{19}$ |

### 4.2. Motor and Control System

The hoisting mechanism use a YZR-type wound three-phase asynchronous motor. As to solve the voltage equation, flux equation, torque equation and motion equation of the motor, the motor model is usually converted from the three-phase stationary A, B, and C coordinate systems to the two-phase synchronous rotating coordinate system, as shown in Equations (8)–(10) [27], through Clarke coordinates and a Park transformation.

Voltage equation:

$$\begin{cases} u_{sd} = R_s i_{sd} + p\psi_{sd} - \psi_{sq}\omega_1 \\ u_{sq} = R_s i_{sq} + p\psi_{sq} + \psi_{sd}\omega_1 \\ u_{rd} = R_r i_{rd} + p\psi_{rd} - \psi_{rq}(\omega_1 - \omega_s) \\ u_{rq} = R_r i_{rq} + p\psi_{rq} - \psi_{rd}(\omega_1 - \omega_s) \end{cases} \tag{8}$$

Flux equation:

$$\begin{cases} \psi_{sd} = L_s i_{sd} + L_m i_{rd} \\ \psi_{sq} = L_s i_{sq} + L_m i_{rq} \\ \psi_{rd} = L_m i_{sd} + L_r i_{rd} \\ \psi_{q2} = L_m i_{sq} + L_r i_{rq} \end{cases} \tag{9}$$

Torque equation:

$$T_e = n_p L_m \left( i_{sq} i_{rd} - i_{rq} i_{sd} \right) \tag{10}$$

Motion equation:

$$T_e = T_m + \frac{J_e}{n_p} \cdot \frac{d\omega}{dt} \tag{11}$$

where $u_{sd}$, $u_{sq}$, $u_{rd}$, and $u_{rq}$ are the components of the stator voltage and rotor voltage on the $d$ and $q$ coordinate axes, respectively. $i_{sd}$, $i_{sq}$, $i_{rd}$ and $i_{rq}$ are the components of the stator current and rotor current on the $d$ and $q$ coordinate axes, respectively. $R_s$ and $R_r$ are the winding resistances of the stator and rotor, respectively. $\psi_{sd}$, $\psi_{sq}$, $\psi_{rd}$ and $\psi_{rq}$ are the components of the stator and rotor flux linkages in the $d$ and $q$ coordinate axes, respectively. $L_s$, $L_r$ and $L_m$ are the inductance of stator and rotor and the mutual inductance between the stator and rotor. $\omega_1$ is the rotational angular velocity of the $d$, $q$ coordinate axis system, and, where $f$ is the AC power frequency for the motor. $\omega_s$ is the slip velocity, and $\omega_s = (\omega_1 - \omega)$, where $\omega$ is the rotor speed. $J_e$ is the rotational inertia of the unit. $n_p$ is the polar number of the motor. $T_m$ is the motor load resistance torque.

To realize the stable operation of the hoisting mechanism, it is necessary to control the motor by the AC speed control system. A crane is a special equipment, and its motor control system typically uses a stable and reliable thyristor stator regulating voltage to achieve speed regulation. The torque equation [28] of the rotor is:

$$T_e = \frac{3 n_p U_s^2 R_r' / s}{\omega_1 \left[ (R_s + R_r'/s)^2 + \omega_1^2 (L_{ls} + L_{lr}')^2 \right]} \tag{12}$$

where $n_p$ is the pole pair of the motor, $U_s$ and $\omega_1$ are the stator phase voltage and supply angular frequency of the motor, $s$ is the slip ratio, $R_s$ and $R_r'$ are the resistance of each phase of the stator and the resistance of each phase of the rotor is converted to the stator side, $L_{ls}$ and $L_{lr}'$ are the leakage inductance of each phase of the stator and the leakage inductance of each phase of the rotor is converted to the stator side.

### 4.3. Mass-Stiffness-Damping a Model of Transmission Mechanisms

Usually, the dynamic equation of transmission mechanisms can be written as the differential equation in the form matrix shown in Equation (13).

$$M\ddot{q} + C\dot{q} + Kq = Q(q, \dot{q}, t) \tag{13}$$

where $M$, $C$ and $K$ are the mass matrix, damping matrix and stiffness matrix of the system, respectively; $q$, $\dot{q}$ and $\ddot{q}$ are the displacement matrix, velocity matrix and acceleration matrix of the system in generalized coordinates, respectively; and $Q(q, \dot{q}, t)$ is the generalized force matrix of the system, which is composed of the gravity and external moment of each equivalent mass unit of the crane.

According to the kinetic model and parameters in Figure 12, the components of Equation (13) can be expressed in matrix form as follows.

$q$, the generalized coordinate displacement matrix, can be expressed as:

$$q_{21\times1} = [y_1, y_l, y_r, y_4, y_7, y_8, \phi_1, \phi_2, \cdots, \phi_{15}]^T \tag{14}$$

$M$, the mass matrix, is a $21\times21$ diagonal matrix that can be expressed as:

$$M = \begin{bmatrix} m_{6\times6} & \mathbf{0} \\ \mathbf{0} & J_{15\times15} \end{bmatrix} \tag{15}$$

where $m_{6\times6}$ is a diagonal matrix of equivalent masses, as shown in Formula (16). $J_{15\times15}$ is the matrix of the rotational inertia, as shown in Formula (17), and $J_h$ is a $6 \times 6$ rotational inertia matrix, as shown in Formula (18).

$$m_{6\times6} = \begin{bmatrix} m_1 & 0 & & \cdots & & 0 \\ 0 & m_j & & & & \\ & & m_j & \ddots & & \vdots \\ \vdots & & \ddots & m_4 & & \\ & & & & m_7 & 0 \\ 0 & & \cdots & & 0 & m_8 \end{bmatrix} \tag{16}$$

$$J_{15\times15} = \begin{bmatrix} J_1 & 0 & \cdots & & 0 \\ 0 & J_2 & & & \\ \vdots & & \ddots & & \vdots \\ & & & J_9 & 0 \\ 0 & \cdots & & 0 & J_h \end{bmatrix} \tag{17}$$

$$J_h = diag(J_h, J_h, J_h, J_h, J_h, J_h) \tag{18}$$

$C$, the damping matrix is a $21\times21$ symmetric matrix that can be expressed by Equation (19), and $C = C^T$.

$$C = \begin{bmatrix} C_{11} & C_{12} & C_{13} & 0 \\ C_{21} & C_{22} & C_{23} & C_{24} \\ C_{31} & C_{32} & C_{33} & C_{34} \\ 0 & C_{42} & C_{43} & C_{44} \end{bmatrix} \tag{19}$$

where $C_{11}$, $C_{21}$, $C_{22}$, $C_{31}$, $C_{32}$, $C_{33}$, $C_{42}$, $C_{43}$, and $C_{44}$ can be expressed by Equations (20)–(28).

$$C_{11} = \begin{bmatrix} (c_1 + 4c_l + 4c_r) & -4c_l & -4c_r & 0 & 0 & -c_1 \\ -4c_l & (4c_l + c_{18}) & 0 & -c_{18} & 0 & 0 \\ -4c_r & 0 & (4c_r + c_{19}) & 0 & -c_{19} & 0 \\ 0 & -c_{18} & 0 & c_{18} & 0 & 0 \\ 0 & 0 & -c_{19} & 0 & c_{19} & 0 \\ -c_1 & 0 & 0 & 0 & 0 & (c_{20} + c_1) \end{bmatrix}_{6\times6} \tag{20}$$

$$C_{21} = \begin{bmatrix} 0 & & \cdots & & 0 \\ 0 & & & & \\ 0 & & \ddots & & \vdots \\ 2c_l r_4 & 2c_l r_4 & & & \\ 0 & 0 & 0 & \cdots & 0 \end{bmatrix}_{5\times 6} \tag{21}$$

$$C_{22} = \begin{bmatrix} c_2 & -c_2 & 0 & 0 & 0 \\ -c_2 & (c_2+c_3) & -c_3 & 0 & 0 \\ 0 & -c_3 & \left(c_3+\frac{1}{n^2}c_4\right) & -\frac{c_4}{n} & 0 \\ 0 & 0 & -\frac{c_4}{n} & (c_4+c_5+2c_l r_4{}^2) & -c_5 \\ 0 & 0 & 0 & -c_5 & (c_5+c_9) \end{bmatrix}_{5\times 5} \tag{22}$$

$$C_{31} = \begin{bmatrix} 0 & 0 & \cdots & \cdots & 0 \\ 0 & 0 & \vdots & & \vdots \\ 0 & 0 & \cdots & \cdots & 0 \\ 2c_r r_9 & 0 & -2c_r r_9 & 0 & 0 & 0 \end{bmatrix}_{4\times 6} \tag{23}$$

$$C_{32} = \begin{bmatrix} 0 & & \cdots & 0 & 0 \\ \vdots & \ddots & & & 0 \\ & & \ddots & \vdots & 0 \\ 0 & \cdots & & 0 & -c_9 \end{bmatrix}_{4\times 5} \tag{24}$$

$$C_{33} = \begin{bmatrix} c_6 & -c_6 & 0 & 0 \\ -c_6 & (c_6+c_7) & -c_7 & 0 \\ 0 & -c_7 & \left(c_7+\frac{1}{n^2}c_8\right) & -\frac{c_8}{n} \\ 0 & 0 & -\frac{c_8}{n} & (c_8+c_9+2c_r r_9{}^2) \end{bmatrix}_{4\times 4} \tag{25}$$

$$C_{42} = \begin{bmatrix} 0 & \cdots & 0 & c_l r_h r_4 & 0 \\ & \ddots & 0 & -c_l r_h r_4 & 0 \\ \vdots & & 0 & 0 & 0 \\ & & & \ddots & \vdots \\ 0 & & \cdots & & 0 \end{bmatrix}_{6\times 5} \tag{26}$$

$$C_{43} = \begin{bmatrix} 0 & \cdots & 0 & 0 \\ & \ddots & \vdots & 0 \\ \vdots & & 0 & c_r r_h r_9 \\ 0 & & 0 & -c_r r_h r_9 \\ & \ddots & \vdots & 0 \\ 0 & \cdots & 0 & 0 \end{bmatrix}_{6\times 4} \tag{27}$$

$$C_{44} = \begin{bmatrix} 2c_l r_h{}^2 & 0 & 0 & 0 & 0 & c_l r_h{}^2 \\ 0 & 2c_l r_h{}^2 & 0 & 0 & 0 & c_l r_h{}^2 \\ 0 & 0 & 2c_r r_h{}^2 & 0 & c_r r_h{}^2 & 0 \\ 0 & 0 & 0 & 2c_r r_h{}^2 & c_r r_h{}^2 & 0 \\ 0 & 0 & c_r r_h{}^2 & c_r r_h{}^2 & 2c_r r_h{}^2 & 0 \\ c_l r_h{}^2 & c_l r_h{}^2 & 0 & 0 & 0 & 2c_l r_h{}^2 \end{bmatrix}_{6\times 6} \tag{28}$$

Because, $C$, the damping matrix, is a symmetric matrix; thus, $C_{12} = C_{21}{}^T$, $C_{13} = C_{31}{}^T$, $C_{23} = C_{32}{}^T$, $C_{24} = C_{42}{}^T$, and $C_{34} = C_{43}{}^T$.

$K$, the stiffness matrix is a $21 \times 21$ symmetric matrix that can be expressed by Equation (29), and $K = K^T$.

$$CK = \begin{bmatrix} K_{11} & K_{12} & K_{13} & 0 \\ K_{21} & K_{22} & K_{23} & K_{24} \\ K_{31} & K_{32} & K_{33} & K_{34} \\ 0 & K_{42} & K_{43} & K_{44} \end{bmatrix} \tag{29}$$

where $K_{11}, K_{21}, K_{22}, K_{31}, K_{32}, K_{33}, K_{42}, K_{43}$, and $K_{44}$ can be expressed by Equations (30)–(38).

$$K_{11} = \begin{bmatrix} (k_1 + 4k_l + 4k_r) & -4k_l & -4k_r & 0 & 0 & -k_1 \\ -4k_l & (4k_l + k_{18}) & 0 & -k_{18} & 0 & 0 \\ -4k_r & 0 & (4k_r + k_{19}) & 0 & -k_{19} & 0 \\ 0 & -k_{18} & 0 & k_{18} & 0 & 0 \\ 0 & 0 & -k_{19} & 0 & k_{19} & 0 \\ -k_1 & 0 & 0 & 0 & 0 & (k_{20} + k_1) \end{bmatrix}_{6 \times 6} \tag{30}$$

$$K_{21} = \begin{bmatrix} 0 & & \cdots & & & 0 \\ 0 & & & & & \\ 0 & & & \ddots & & \vdots \\ 2k_l r_4 & 2k_l r_4 & & & & \\ 0 & 0 & 0 & \cdots & & 0 \end{bmatrix}_{5 \times 6} \tag{31}$$

$$K_{22} = \begin{bmatrix} k_2 & -k_2 & 0 & 0 & 0 \\ -k_2 & (k_2 + k_3) & -k_3 & 0 & 0 \\ 0 & -k_3 & \left(k_3 + \frac{1}{n^2}k_4\right) & -\frac{k_4}{n} & 0 \\ 0 & 0 & -\frac{k_4}{n} & (k_4 + k_5 + 2k_l r_4{}^2) & -k_5 \\ 0 & 0 & 0 & -k_5 & (k_5 + k_9) \end{bmatrix}_{5 \times 5} \tag{32}$$

$$K_{31} = \begin{bmatrix} 0 & 0 & \cdots & \cdots & & 0 \\ 0 & 0 & \vdots & & & \vdots \\ 0 & 0 & \cdots & \cdots & & 0 \\ 2k_r r_9 & 0 & -2k_r r_9 & 0 & 0 & 0 \end{bmatrix}_{4 \times 6} \tag{33}$$

$$K_{32} = \begin{bmatrix} 0 & & \cdots & 0 & 0 \\ \vdots & \ddots & & & 0 \\ & & \ddots & \vdots & 0 \\ 0 & \cdots & & 0 & -k_9 \end{bmatrix}_{4 \times 5} \tag{34}$$

$$K_{33} = \begin{bmatrix} k_6 & -k_6 & 0 & 0 \\ -k_6 & (k_6 + k_7) & -k_7 & 0 \\ 0 & -k_7 & \left(k_7 + \frac{1}{n^2}k_8\right) & -\frac{k_8}{n} \\ 0 & 0 & -\frac{k_8}{n} & (k_8 + k_9 + 2k_r r_9{}^2) \end{bmatrix}_{4 \times 4} \tag{35}$$

$$K_{42} = \begin{bmatrix} 0 & \cdots & 0 & k_l r_h r_4 & 0 \\ & \ddots & 0 & -k_l r_h r_4 & 0 \\ \vdots & & 0 & 0 & 0 \\ & & & \ddots & \vdots \\ 0 & & \cdots & & 0 \end{bmatrix}_{6 \times 5} \tag{36}$$

$$\mathbf{K}_{43} = \begin{bmatrix} 0 & \cdots & 0 & 0 \\ & \ddots & \vdots & 0 \\ \vdots & & 0 & k_r r_h r_9 \\ 0 & & 0 & -k_r r_h r_9 \\ & \ddots & \vdots & 0 \\ 0 & \cdots & 0 & 0 \end{bmatrix}_{6\times 4} \tag{37}$$

$$\mathbf{K}_{44} = \begin{bmatrix} 2k_l r_h{}^2 & 0 & 0 & 0 & 0 & k_l r_h{}^2 \\ 0 & 2k_l r_h{}^2 & 0 & 0 & 0 & k_l r_h{}^2 \\ 0 & 0 & 2k_r r_h{}^2 & 0 & k_r r_h{}^2 & 0 \\ 0 & 0 & 0 & 2k_r r_h{}^2 & k_r r_h{}^2 & 0 \\ 0 & 0 & k_r r_h{}^2 & k_r r_h{}^2 & 2k_r r_h{}^2 & 0 \\ k_l r_h{}^2 & k_l r_h{}^2 & 0 & 0 & 0 & 2k_l r_h{}^2 \end{bmatrix}_{6\times 6} \tag{38}$$

Since $\mathbf{K}$ is a symmetric matrix. Thus $\mathbf{K}_{12} = \mathbf{K}_{21}{}^T$, $\mathbf{K}_{13} = \mathbf{K}_{31}{}^T$, $\mathbf{K}_{23} = \mathbf{K}_{32}{}^T$, $\mathbf{K}_{24} = \mathbf{K}_{42}{}^T$, $\mathbf{K}_{34} = \mathbf{K}_{43}{}^T$.

$\mathbf{Q}$, the generalized force matrix, is defined in Equation (39).

$$\mathbf{Q} = \begin{bmatrix} \mathbf{O}_{1\times 3} & -m_4 g & -m_7 g & 0 & T_{e1} & -T_2 & \mathbf{O}_{1\times 3} & T_{e6} & -T_7 & \mathbf{O}_{1\times 8} \end{bmatrix}^T \tag{39}$$

where $T_{e1}$ and $T_{e6}$ are the driving torque of motors 1 and 6, respectively, and $T_2$ and $T_7$ are the braking torque of brakes 2 and 7, respectively.

According to the dynamic model of the hoisting mechanism, the load torque of the motor can be obtained as follows:

$$T_{m1} = c\left(\dot{\theta}_1 - \dot{\theta}_2\right) + k(\theta_1 - \theta_2) \tag{40}$$

$$T_{m6} = c\left(\dot{\theta}_6 - \dot{\theta}_7\right) + k(\theta_6 - \theta_7) \tag{41}$$

where $c$ is the damping coefficient, $\theta$ and $\dot{\theta}$ are the rotation angle and the rotational speed of the rotor, respectively, and $T_{m1}$ and $T_{m6}$ are the load torque of the motor 1 and motor 6, respectively.

According to the coupling relationship between the motor control model and the mass-stiffness-damping model of the hoisting mechanism, the following equations can be obtained as:

$$\omega_{r1} = \dot{\theta}_1 \tag{42}$$

$$\omega_{r6} = \dot{\theta}_6 \tag{43}$$

Since both the motor driving torque and the brake braking torque are the result of the coupling between the motor and the mechanical structure, they are not constant values. It is difficult to describe these torques by a single time-varying function. The motor driving torque is the real time output signal of the motor, and the brake braking torque is the real time resistance torque after the brake start to act. Therefore, as to accurately reflect the dynamic response of the hoisting mechanism under various working conditions, it is necessary to simultaneously solve for the motor driving torque and the brake braking torque.

### 4.4. Calculation of the Structural Parameters of the Hoisting Mechanism

To solve the abovementioned dynamic Equations, it is necessary to first solve for the parameters in Table 8, such as the equivalent mass, equivalent moment, damping and stiffness of the main beam.

### 4.4.1. Calculation of Equivalent Mass of the Main Girder

Because of the long length of the main girder, the mass distributed in the direction of the length of the main beam is equivalent to the position of the lifting trolley. When the lifting trolley is at a certain distance from the left end of the main girder, the mass of the main beam is equivalent to:

$$m_8 = \frac{3L^4 m}{\pi^4 a^2 (L-a)^2} \tag{44}$$

where $a$ is the distance between the hoisting trolley and the left end of the main girder, $L$ is the length of the main girder, and m is the total mass of the main beam steel structure.

### 4.4.2. Calculation of the Equivalent Moment of Inertia

When the hoisting mechanism is running, the rotating shafts of brakes, motors, reducers, drums and other structural parts were not on the same axis. Therefore, it is necessary to convert the inertia moment of each component to the power takeoff shaft.

The kinetic energy of the moving parts in the transmission system is:

$$E_1 = \frac{1}{2} \sum_{i=1}^{m} J_i \omega_i^2 + \frac{1}{2} \sum_{j=1}^{n} m_j v_j^2 \tag{45}$$

Assuming that the equivalent inertia moment of load is $J_L$, the equivalent kinetic energy on the motor shaft is:

$$E_2 = \frac{1}{2} J_L \omega_L^2 \tag{46}$$

From the law of energy conservation law, $E_1 = E_2$, the equivalent inertia moment of load is:

$$J_L = \sum_{i=1}^{m} J_i \left( \frac{\omega_i}{\omega_L} \right)^2 + \sum_{j=1}^{n} m_j \left( \frac{v_j}{\omega_L} \right)^2 \tag{47}$$

### 4.4.3. Calculation of the Damping Coefficient

According to viscous damping theory, the damping force is proportional to the moving speed, and the absolute value of the ratio of the damping force to the speed is the damping coefficient. Additionally, the damping factor is usually calculated by multiplying the damping ratio and the critical damping factor. That is, the damping coefficient of the part can be obtained as:

$$c_i = \zeta c_{ci} = \zeta \times 2\sqrt{k_i m_i} (i = 1, 2 \ldots 20) \tag{48}$$

where $c_i$, $\zeta$, $c_{ci}$, $k_i$ and $mi$ are the actual damping coefficient, damping ratio, critical damping coefficient, stiffness and mass of the part, respectively. The damping ratio $\zeta$ is taken as 0.1.

### 4.4.4. Calculation of Stiffness Coefficient

In the hoisting mechanism, the calculation of stiffness mainly involves the main girder, the rotating shaft and the wire rope.

The stiffness of main girder is calculated by using the equivalent calculation method. When the lifting trolley is at a certain distance from the left end of the main girder, the stiffness of the main girder can be obtained as:

$$k_8 = \frac{3EIL}{a^2 (L-a)^2} \tag{49}$$

where $E$ is the elastic modulus of the main girder and $I$ is the inertia moment of cross-sectional.

The rotational stiffness of the rotating shaft is calculated using Equation (50):

$$k_t = \frac{T}{\varphi} = \frac{GI_p}{l} \tag{50}$$

where $T$ is the torque acting on the rotating shaft, $\varphi$ is the torsion angle of the rotating shaft, $G$ is the shear modulus of elasticity of the torsion shaft, and $I_p$ is the length of the torsion shaft.

The stiffness of the wire rope is related to its length, and the stiffness of a single wire rope can be calculated using Equation (51).

$$k_i = \frac{E_z A}{L_0 - \sum y_i} \tag{51}$$

where $E_z$ is the elastic coefficient of the wire rope, $E_z = 110 \times 109$ N$^2$/m, $A$ is the area of the wire rope cross-section, $A = 6.154 \times 10^{-4}$ m$^2$, $L_0$ is the initial length of the wire rope, $L_0 = 6$ m, and $\sum y_i$ is the displacement of the shortened length of the unilateral wire rope.

According to the above method, the values of the parameters in Table 7 were calculated, as shown in Table 10.

**Table 10.** The values of the parameters in Table 7.

| Parameters | Value | Unit | Parameters | Value | Unit |
|---|---|---|---|---|---|
| $m_1$ | 44,800 | kg | $m_j$ | 3500 | kg |
| $m_4$ | 25,000 | kg | $m_7$ | 25,000 | kg |
| $m_8$ | 90,000 | kg | $J_2, J_7$ | 0.049 | kg·m$^2$ |
| $J_1, J_6$ | 7.22 | kg·m$^2$ | $J_4, J_9$ | 2158 | kg·m$^2$ |
| $J_3, J_8$ | 0.1633 | kg·m$^2$ | $J_{10} \sim J_{15}$ | 0.04 | kg·m$^2$ |
| $J_5$ | 26.45 | kg·m$^2$ | $k_1$ | $8 \times 106$ | N/m |
| $k_{20}$ | $9 \times 106$ | N/m | $k_3, k_7$ | $4.26 \times 104$ | N/m |
| $k_2, k_6$ | $1.26 \times 104$ | N/m | $k_4, k_8$ | $2 \times 105$ | N/m |
| $k_r, k_l$ | $4.1 \times 105$ | N/m | $k_5, k_9$ | $2 \times 105$ | N/m |
| $k_{18}, k_{19}$ | $2.5 \times 105$ | N/m | $c_1$ | 80,000 | N·s/m |
| $c_{20}$ | 90,000 | N·s/m | $c_3, c_7$ | 5000 | N·s/m |
| $c_2, c_6$ | 5000 | N·s/m | $c_4, c_8$ | 5000 | N·s/m |
| $c_r, c_l$ | 4080 | N·s/m | $c_5, c_9$ | 2000 | N·s/m |
| $r_4, r_9$ | 0.7 | m | $c_{18}, c_{19}$ | $6.5 \times 104$ | N·s/m |
| $r_i$ (i = 10,11, . . . ,15) | 0.09 | m | $N$ | 173.03 | |

Thus, by substituting the system parameters into the mathematical model of the motor system and transmission mechanisms, the dynamic load distribution of the bilaterally driven hoisting mechanism can be solved.

*4.5. Comparison of Dynamic Simulation and Experiment*

According to the procedure shown in Figure 11, the dynamic simulation of the hoisting mechanism under different working conditions was conducted.

Taking the test condition shown in Figure 10a as an example, when the hoisting weight (40.44 t) of the hoisting mechanism was in the process of hoisting and braking in the 1st gear, the torque at measuring point 2 on the load drum shaft was tested and simulated. The simulation and test torques are shown in Figure 13.

Figure 13 shows the torque at point 2 on the spool shaft before or after braking. The vibration amplitude of the simulated torque was larger than it in test, the mean value of the simulated torque was smaller than the tested torque, and the residual torque after braking in simulation was also smaller than it in test. These results may have been related to the assumptions such as ignoring friction in dynamic modeling, and further quantitative analysis was needed.

However, the tested torque and simulated torque periodic oscillated with a period of 13 s during the stable lifting process. Additionally, the tested torque and simulated torque at the measuring point vibrated during braking and retained stable residual torque after braking. These meant that the dynamic responses obtained by the simulation and test were

basically the same. Thus, using the dynamic model to further analyze the fatigue fracture failure of the load drum shaft was credible.

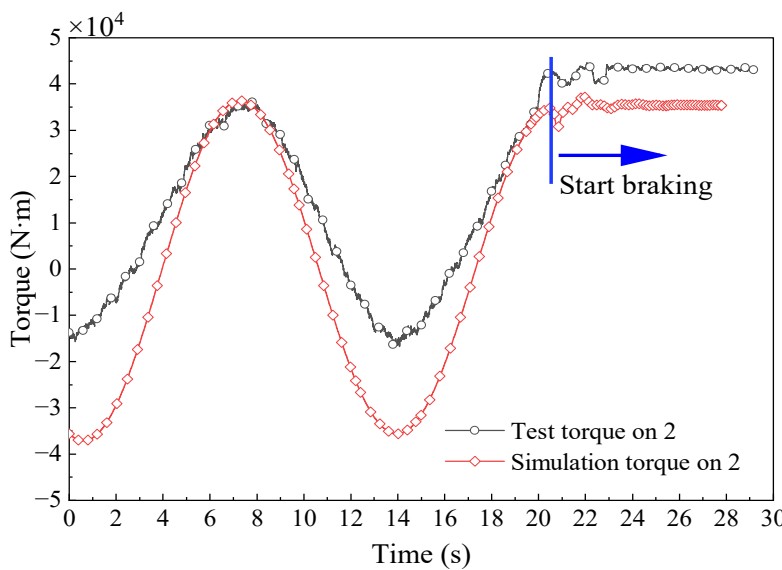

**Figure 13.** Comparison of simulation and experimental data for the torque at test point 2.

## 5. Simulation Results and Application Analysis

### 5.1. Effects of the Motor and Its Controls

According to the mathematical model, the simulation model of the three-phase asynchronous motor and its speed control system was built in the MATLAB/Simulink, as shown in Figure 14. The motor model mainly included the stator module, rotor module, flux linkage module and torque module. The motor speed control module mainly included the thyristor stator speed regulation module with voltage regulation, the rotor speed regulation module with connecting resistance, and the braking control module with energy consumption.

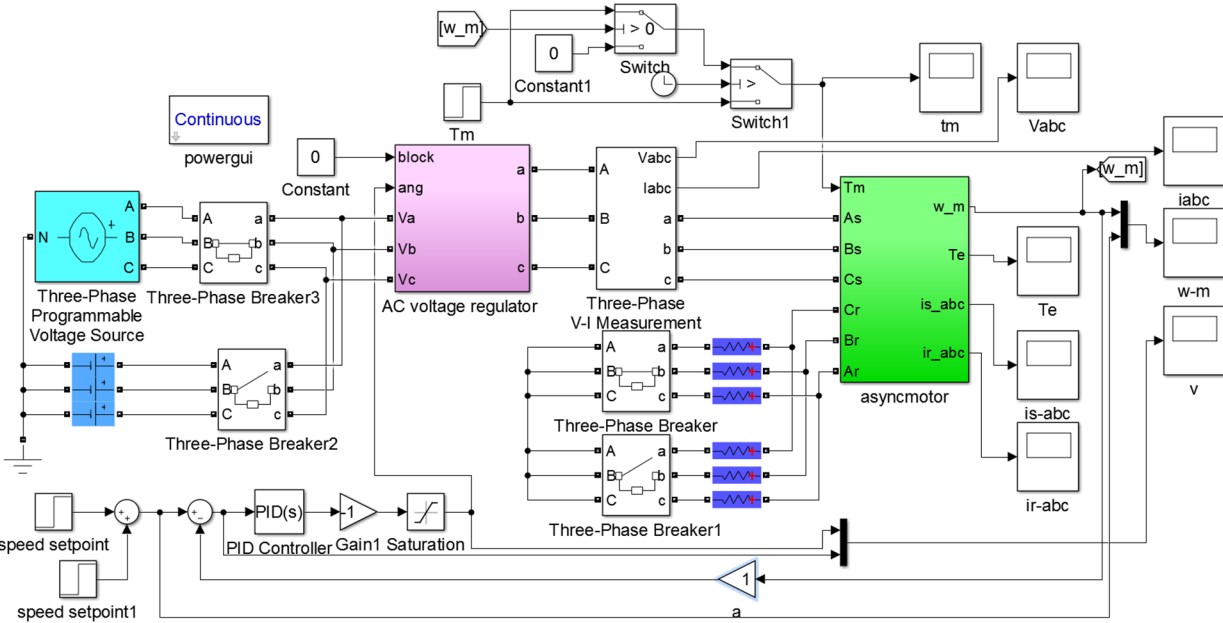

**Figure 14.** Three-phase asynchronous motor and its speed control model.

As to analyze the electromechanical coupling effect more clearly, the load was directly connected to the motor in Section 4.1. The parameters used in the simulation are shown in Table 11.

**Table 11.** Motor performance and its values.

| Performance | Value | Performance | Value |
|---|---|---|---|
| Phase voltage/V | 380 | Rotor resistance/$\Omega$ | 0.027 |
| Power frequency/Hz | 50 | Rotor leakage inductance/H | 0.000462 |
| DC voltage/V | 15 | Mutual inductance/H | 3.6 |
| Rotor resistance at startup/$\Omega$ | 0.873 | Pole pairs | 4 |
| Rotor resistance at steady state/$\Omega$ | 0.209 | Rotor moment/kg·m$^2$ | 7.22 |
| Stator resistance/$\Omega$ | 0.042 | Friction coefficient | 0.0 |
| Stator leakage inductance/H | 0.0000296 | | |

The given motor speed and load torque during the simulation are shown as solid lines in Figures 15 and 16, and the total simulation time was 12 s. When starting, the given motor speed was 15 rad/s, and the load was 300 N m. At 3 s and 5 s, the given speed was increased to 30 rad/s and 75 rad/s, respectively. At 8 s, the speed remained unchanged, and the load was increased to 500 N m. At 10 s, the brake was turned on until the rotor speed dropped to zero, that is, the motor load became zero. The dotted line in Figure 15 shows the rotational speed obtained by the simulation.

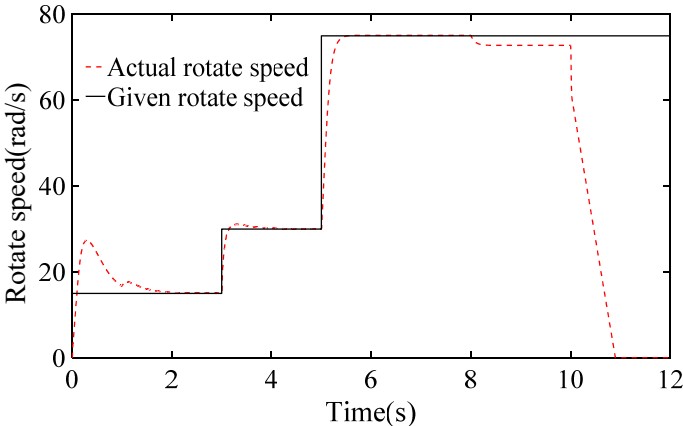

**Figure 15.** Motor given speed and actual speed.

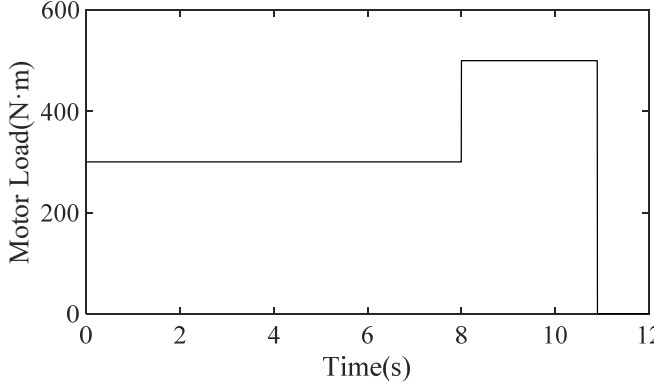

**Figure 16.** The given load torque of motor.

Figure 15 shows that the actual speed of the motor tracked the change in the given speed, and it decreased slightly and stabilized quickly when the load increased. In addition,

when the motor started to brake, the rotate speeds decreased rapidly, and there was a sudden change in acceleration for the electric braking torque was not stable.

Under the given speed and load, as shown in Figures 15 and 16, the current and electromagnetic torque of motor are shown in Figures 17 and 18. A comparison showed that the changes in currents of stator and rotor were synchronous with the changes in electromagnetic torque. Additionally, the three-phase current of the rotor followed the change in the three-phase current of the stator, and due to the load, there was a slip rate between them. The frequency of three-phase current in the rotor was lower than it in the stator.

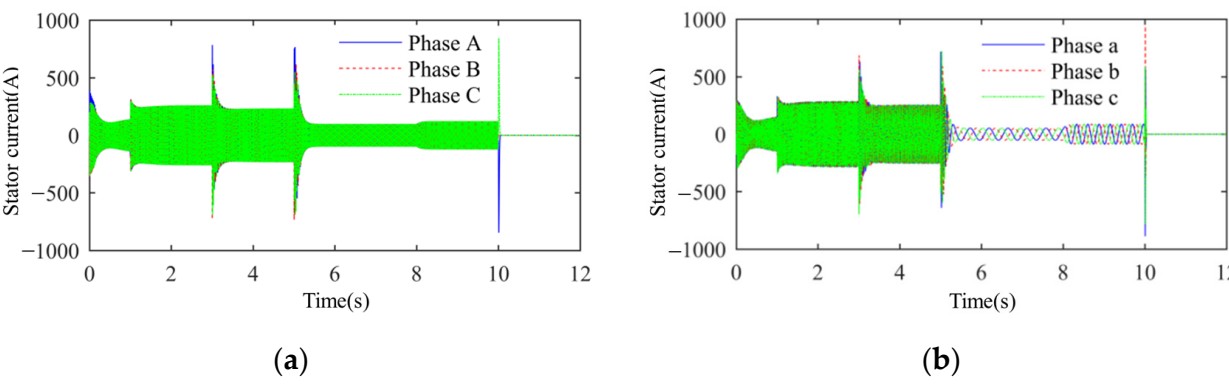

(**a**)  (**b**)

**Figure 17.** Current curves of motor: (**a**) three-phase current of stator; (**b**) three-phase current of rotor.

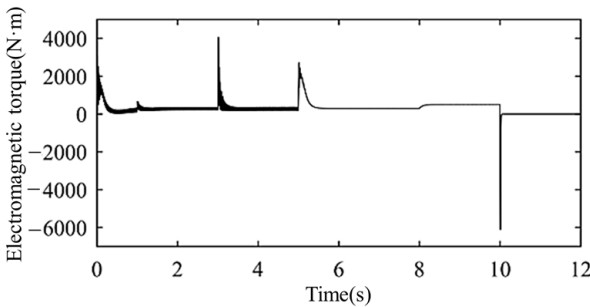

**Figure 18.** Electromagnetic torque of motor.

It is well known that the electromagnetic torque of motor depends on its stator current, which is generated by the stator voltage in the stator winding. In reality, the loads also had influences on the induced electromotive force and induced current on the rotor, which finally affected the stator current. This was the principle of coupling between the motor and the mechanical load, which made the electromagnetic torque of motor undergo complex dynamic changes.

Figure 18 shows that in the processes of starting, regulating the speed, braking and changing the load, the motor ensured a relatively stable speed, but the output torque obviously underwent overshoots and fluctuations. This inevitably resulted in large dynamic forces on the mechanical transmission and result in its fatigue fracture, including the load drum shaft.

### 5.2. Effects of Nonsynchronous Control

According to design requirements, the structures of the bilateral drive hoisting mechanism should be symmetrical, and the control should be synchronized. However, the asynchronous start of the motor and the asynchronous braking of the brakes often happened, which could result in the motor towing.

### 5.2.1. Single-motor Towing

If the right motor (motor 6) in Figure 12 was damaged, then, only the left motor (motor 1) would drive the entire hoisting mechanism at 1st gear speed to raise the object, and the operation time would be 20 s. The variations in the parameters of the hoisting mechanism from the simulation are shown in Figures 19–22.

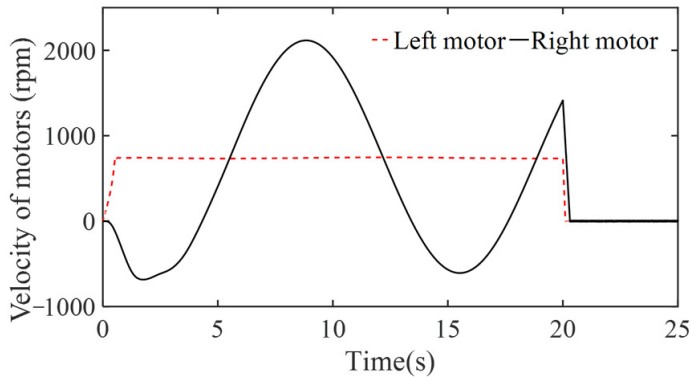

**Figure 19.** Speed of the motor shaft when driving a single motor.

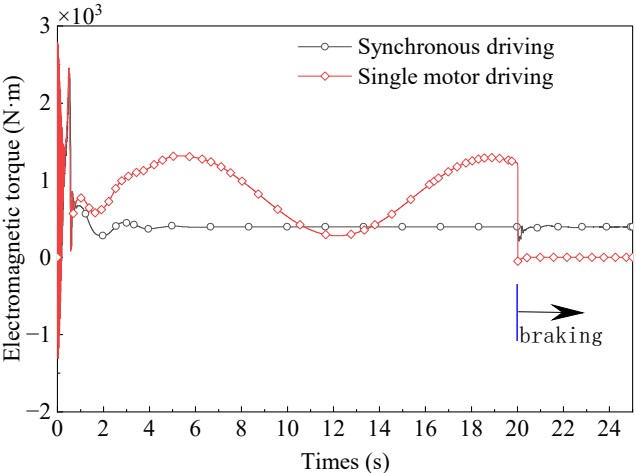

**Figure 20.** Electromagnetic torque comparison.

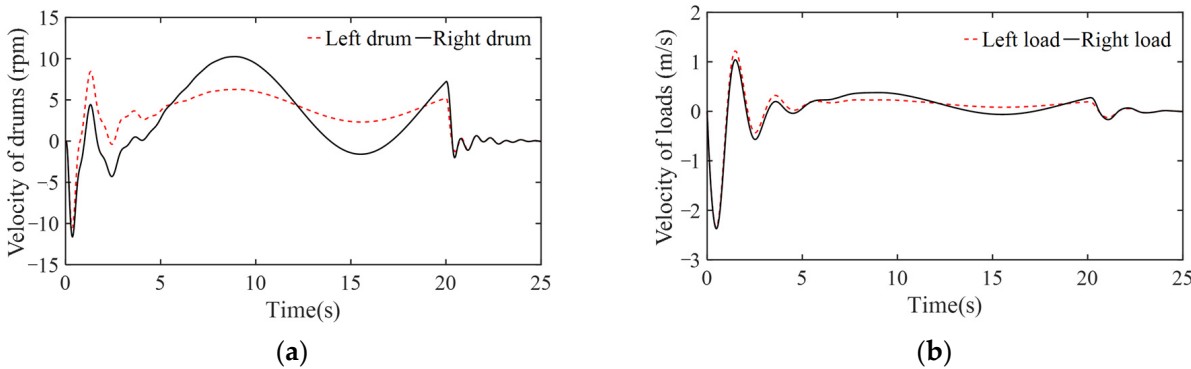

**Figure 21.** Drum speed and lifting speed when a single motor is driving: (**a**) left and right drum speeds; (**b**) lifting speed.

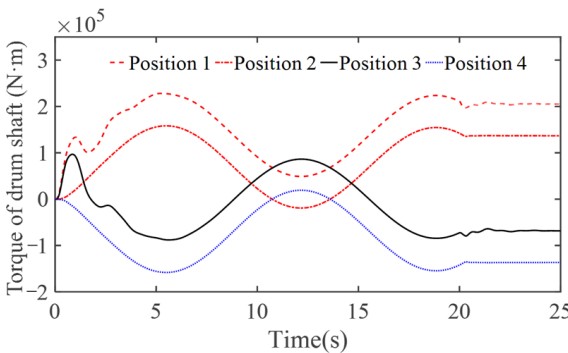

**Figure 22.** Torque of drum shaft when a single motor is driving.

As shown in Figure 19, under the motor speed control system, the speed of the left motor changed smoothly. However, the right motor 6 was towed, and its rotational speed oscillated with a period of 13 s. Additionally, the average speed of right motor was equal to that of the left motor.

As shown in Figure 20, the electromagnetic torque of the left motor exhibited overshoot behavior that was basically consistent with the oscillation of the electromagnetic torque during synchronous driving. Additionally, the electromagnetic torque of the left motor was twice the average when the hoisting mechanism was in synchronous drive. In addition, comparison of Figures 19 and 20 shows that the electromagnetic torque of left motor oscillated periodically, with a 90-degree out-of-phase difference from the rotational speed.

As shown in Figures 21 and 22, single-motor towing also affected the dynamic behavior of the two load drums. Figure 21a shows that the rotational speed amplitude of the left load drum, 5 r/min, was less than that of the right load drum, 15 r/min, but their oscillation tendencies were basically the same. The speeds of the two load drums decreased rapidly after braking and stopped after short-term oscillation. It can be concluded from Figure 21b that the rotational speeds of the two load drums oscillated with a period of 13 s after 5 s, and the average values were both 4 r/min, which ensured that the lifting speeds of the left and right sides were basically the same.

As shown in Figure 22, when the left motor towed the right motor, the torque was significantly larger on the input shaft of the left load drum than the input shaft of the right load drum after startup. After 5 s, the torques of the shafts of the left and right drums both fluctuated with a period of 13 s. Additionally, after braking, there were constant residual torques on the shafts of the left and right drums, and the torques on the left and right drum shafts had equal absolute values and opposite directions. Therefore, when a single motor was towed in the bilateral propelled hoisting mechanism, it generated a large alternating load on the drum shaft during starting, stable operation and braking. This was another load source for fatigue fracture of the spool shaft.

However, in actual operation, there is usually no damage to a single motor for a long time. More working conditions lead to motor drag caused by unsynchronized motor startup or brake braking.

### 5.2.2. Motor startup Time Difference

When the hoisting mechanism lifted a heavy object in 1st gear and the startup time difference $\Delta t$ of the two motors was 0.05 s, 0.1 s, 0.3 s, and 0.5 s, the torque of the left and right drum shafts changed, as shown in Figure 23. According to Figure 23, the asynchronous start of the motor generated a large torque on the load drum shaft, and the torque was positively correlated with the time difference, $\Delta t$. When $\Delta t$ decreased from 0.5 s to 0.05 s, the absolute value of the torque decreased from $2.7 \times 10^4$ N·m to $2.5 \times 10^3$ N·m. The torque decreased slowly with running time increased.

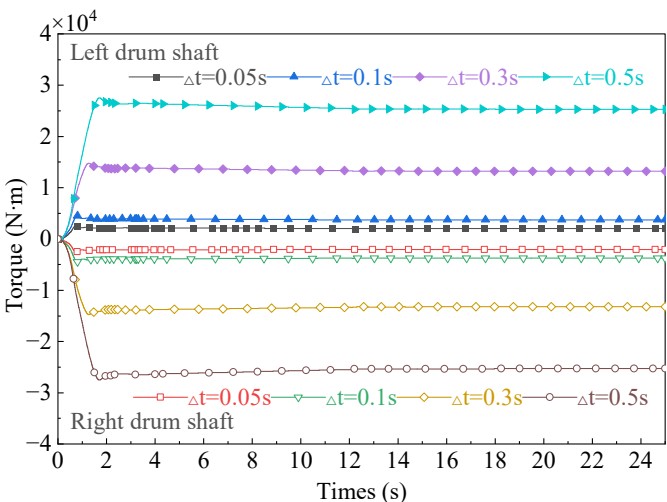

**Figure 23.** Torque on the left and right drum shafts under different power-on time.

### 5.2.3. Brake starting Time Difference

In the design condition, the two motors of the hoisting mechanism were powered on at the same time, heavy objects were lifted in 1st gear for 12 s and then the motors were braked at different times. When the brake starting time difference was 0.05 s, 0.1 s, 0.3 s, and 0.5 s, the torques on the output shafts of the left and right drums are shown in Figure 24. According to Figure 24, when the braking times of the left and right brakes differed, the torque on the output shaft of the load drum suddenly changed and a large residual torque was retained after braking. In addition, the absolute value was equal for the residual torques on the left and right load drum shafts, and both were positively correlated with the brake time difference. When $\Delta t'$ is 0.5 s, the absolute value of the moment reached a maximum of $1.25 \times 10^4$ N·m.

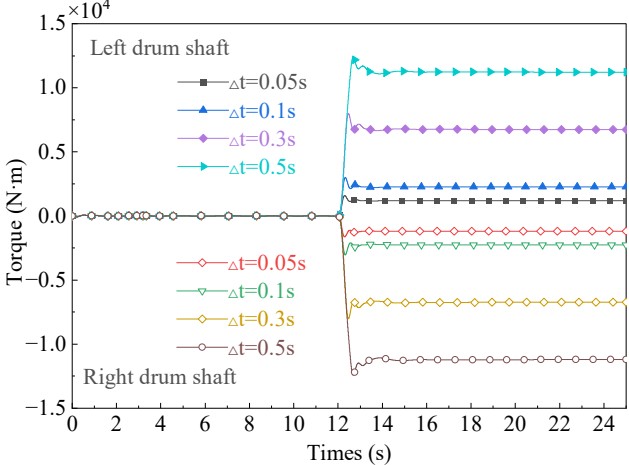

**Figure 24.** Torques on the left and right drum shafts under different brake closing time.

From the above analysis, it can be concluded that although coupling realized the synchronous lifting of hoisting weights, asynchronous control of motors and brakes changed the force distribution in the hoisting mechanism and generated periodic vibrations or constant dynamic loads on the left and right load drum shafts. The instantaneous torque value reached $2 \times 10^5$ N·m, causing shock and fatigue damage to the structure and reducing the service life of the spool shaft.

## 6. Conclusions

Considering fracture morphology and failure factors such as material structure, manufacturing process, and load, the fatigue fracture of the load drum shaft of the bilateral drive casting bridge crane was investigated. According to the test results, dynamic modeling and simulations, the dynamic loading of the crane during operation was a key factor in the fatigue fracture of load drum shafts. The main conclusions are as follows:

(1) The dynamic characteristics testing showed that starting the motor, braking and motor dragging caused by service conditions and control behavior caused the left and right load drum bearings to undergo oscillating torques with the cycles of approximately 13 s. Additionally, starting, shifting, and braking caused sudden changes in the torque on the load drum shaft, affecting the dynamic performance of the hoisting mechanism.

(2) The dynamic simulation results of the hoisting mechanism quantified the influence of the motor speed and load changes on the motor output electromagnetic torque and current changes and the influence of asynchronous control on the torque of the load drum shaft. The results showed that although the coupling can realize synchronous lifting of the left and right hoisting weights, the asynchrony of the left and right speed controls changed the load distribution in the hoisting mechanism and generated additional periodic or constant torques on the left and right load drum shafts. The torque value reached $2 \times 105$ N·m, giving rise to impact and fatigue damage to the structure and reducing the service life of the load drum shaft.

(3) Because several assumptions were made in the formulation of the dynamic simulation model, the simulation results and the test results differed in amplitude, but other dynamic responses were basically the same. Therefore, this dynamic simulation can be used to explain the origins of the fatigue fracture of spool shafts. However, further refinement of the model is required for applications in fatigue-resistant design.

**Author Contributions:** Conceptualization, D.X. and Y.L.; methodology, D.X.; software, Y.L.; validation, Y.Z. and F.X.; formal analysis, D.X. and Y.L.; resources, F.X.; data curation, Y.Z. and F.X.; writing—original draft preparation, Y.L.; writing—review and editing, Y.Z.; project administration, D.X. All authors have read and agreed to the published version of the manuscript.

**Funding:** This research was funded by the National Natural Science Foundation of China under grant number 51975323.

**Conflicts of Interest:** The authors declare no conflict of interest. The funders had no role in the design of the study; in the collection, analyses, or interpretation of data; in the writing of the manuscript; or in the decision to publish the results.

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
