# Peer review of "Experimental Study, Simulation and Analysis of the Fracture Failure of the Drum Shaft of a Casting Bridge Crane"

_electronics, doi:10.3390/electronics11193043_

Round 1

Reviewer 1 Report

The manuscript can be accepted by addressing the following comments.

1) Please rewrite the abstract and add some quantitative findings.

2) The motivation of the study is not clear.

3) Why the torque on the load drum shaft change suddenly by changing the hoisting weight?

4) There are numerous grammatical errors in the manuscript.

Author Response

Thank you for your comments concerning our manuscript. Those comments are valuable and very helpful. We have read through comments carefully and have made corrections. We have uploaded the file of the revised manuscript.

Reviewer 2 Report

The article can be published in its current form. The work clearly presents the described engineering issue and does not require any major revisions.

Author Response

Thanks for your comments. We have made corrections,  please see in the attachment.

No comments to reply.

Reviewer 3 Report

The authors have developed dynamic analytical model to investigate fractures observed in drum shaft of casting bridge crane. The proposed methodology is sound and the model predicts the trends observed, reasonably well. Further calibration of dynamic model and accounting for assumptions will improve the model capability.

The paper is organized properly and well written. However, it is suggested to proof-read the paper to eliminate minor typos, for example, line 199, subsection 3.2 'Test Ssystem Setup'. In addition, all the images need to be enlarged for better clarity.

Can authors comment about simulating this problem using finite element analysis? What are the challenges involved and why finite element analysis wasn't conducted to investigate these failures?

Finally, the scope of the paper seems to be more aligned with MDPI Machines journal instead of MDPI Electronics journal.

Author Response

(The authors gave the same response as above.)

Round 2

Reviewer 3 Report

Dear Authors,

Thank you for you response.